# Determinants of Quality of Life (QoL) in Female Caregivers in Elderly Care Facilities in Korea

**DOI:** 10.3390/bs14010053

**Published:** 2024-01-15

**Authors:** Hee-Kyung Kim, Hye-Suk Oh

**Affiliations:** Department of Nursing, Kongju National University, Gongju 32588, Republic of Korea; d20230184@smail.kongju.ac.kr

**Keywords:** job stress, depression, fatigue, self-efficacy, interpersonal relationship, quality of life

## Abstract

Background: The purpose of this study was to analyze the effects of general characteristics, fatigue, depression, self-efficacy, job stress and interpersonal relationships on the quality of life (QoL) of caregivers in nursing hospitals and use them as basic data for intervention programs to improve the quality of life of caregivers. Methods: The participants in the study were 137 caregivers, aged 52–76, who were actively working in nursing hospitals. Data were collected from caregivers by visiting 9 hospitals in 6 cities, with a questionnaire of fatigue, depression, self-efficacy, job stress, interpersonal relationship, quality of life. Results: Age, marriage, marital satisfaction, education, education experience of QoL, monthly income, perceived economic status, hobby or leisure activity, and number of disease showed differences in the degree of QoL at a statistically significant level. In stage 1, economic status (β = −0.18, *p* = 0.033) and hobby or leisure activity (β = 0.19, *p* = 0.025) were influencing factors (F = 4.58, *p* < 0.001). In stage 2, monthly income (β = −0.19, *p* = 0.034) and perceived economic status (β = −0.18, *p* = 0.035) were influencing factors. In stage 3, age (β = −2.80, *p* = 0.006), perceived economic status (β = −2.41, *p* = 0.017), self-efficacy (β = 3.19, *p* = 0.002) and interpersonal relationship (β = 7.12, *p* < 0.001) were influencing factors which showed 61.5% explanatory power (F = 12.88, *p* < 0.001). Since the subject’s fatigue, depression, and stress did not affect the quality of life, further research is needed. Conclusions: In order to improve the quality of life of caregivers, it would be necessary to develop interventions for raising their self-efficacy and interpersonal relationship by considering their degree of economic status, hobby or leisure activity, monthly income, and age.

## 1. Introduction

Most caregivers in Korean nursing hospitals are middle-aged women over 41 and elderly women over 65, who are not well-off at home. The working environment includes providing services for the elderly, who are often physically weak or working in two or three shifts. This situation inevitably leads to a poor personal quality of life due to various and excessive stress [1]. Quality of life is the feeling of satisfaction derived from the overall conditions that make an individual’s life worthwhile and rewarding. This satisfaction results from the interaction of social conditions, institutions, and social member relationships [2]. Given that the quality of life of a caregiver is closely linked to the quality of care and welfare services for the elderly, it becomes essential to enhance the quality of life of caregivers to improve the quality of care services for the elderly [1]. 

A nursing hospital is defined as a medical institution with 30 or more care beds, providing medical care to patients who require long-term hospitalization [3]. These institutions treat and nurse patients who find it difficult to receive care at home or other residential facilities due to severe conditions, dementia, stroke, or chronic age-related diseases. The caregivers in these places work under a job execution system distinguished by profession and rank, taking directions from other professionals. Most caregivers are women, whose empathetic, meticulous, and relationship-focused attributes are advantageous in Republic of Korea. However, there is a lot of job-related stress because of unsafe working conditions/environments, heavy workload, interpersonal conflicts at work, role ambiguity, job insecurity and bi-dayly work, caregivers of workforce can feel uncomfortable, pressured, tense, and conflicted. As a result, the quality of life of care providers is reduced, and they are exhausted and ultimately considered to change jobs [1,4,5]. Conversely, nursing caregivers experience a high quality of life when providing care for vulnerable elderly individuals in nursing hospitals. This is achieved through accurately identifying the needs of patients, delivering appropriate care, and effectively managing various situations [6]. Therefore, improving the caregiving skills of caregivers, fostering positive interrelationships, and enhancing environmental conditions in the workplace can potentially enhance the quality of life for caregivers.

Caregivers affiliated with nursing hospitals in Korea consist of nursing aides and caregivers, and they constitute a significantly high proportion among long-term care institution workers [7]. They directly provide services to meet the needs of vulnerable elderly patients with diseases, such as dementia and stroke, who find it challenging to perform independent daily life activities, and play a critical role in patient care and transfer [8]. In reality, the number of nurses in Korean nursing hospitals is limited compared to the number of patients, so they cannot care for every patient daily. Since guardians also need to maintain their livelihood and cannot provide care during 24 h, there is a system in place where trained caregivers from hospitals carry out such duties. Institutions where caregivers, comprised of nursing aides and caregivers, operate include nursing hospitals, care centers, etc. Especially in nursing hospitals where patients with severe conditions are admitted, caregivers perform various crucial roles such as observer, communicator, counselor, transporter, advocate, and motivator [4,9]. Therefore, it is necessary to review factors to improve the quality of life of care providers in nursing hospitals.

Meanwhile, one can consider the theory on quality of life by Ferrans et al. [10]. This is a model that integrates both biomedical and sociological perspectives, considering not only physical health but also overall aspects of life. The quality of life elements are evaluated based on biological functions, symptoms, functional status, health perception, personal, and environmental characteristics, and their causal relationships. Therefore, based on the research results related to the quality of life of caregivers in nursing hospitals and the model of Ferrans et al. [10], this study included fatigue as a factor in the biological function, functional status, and general health perception of caregivers. Symptoms are mostly represented by the variable of depression, which is more common in women. As these are negative factors and have relevance to the quality of life, efforts to reduce fatigue and depression are necessary. Furthermore, job-related stress can adversely affect the quality of life, so the variable of job stress should be considered and its level reduced. Mediating and buffering factors include personal characteristics like self-efficacy [11] and environmental characteristics like interpersonal relationships [12]. Enhancing caregivers’ self-efficacy and establishing harmonious human relationships can facilitate their job performance and improve the quality of life [11,12].

Firstly, fatigue in office workers is typically caused by excessive mental and physical labor. In the case of nursing caregivers, the inherently labor-intensive nature of their work targeting humans adds both physical and mental fatigue, posing a threat to their health [13]. Given that caregivers, predominantly middle-aged women over 41 and elderly women over 65, exhibit physical, psychological, and social characteristics, they are particularly sensitive to fatigue [14].

Moreover, the repetitive physical tasks involved in caregiving, such as changing body positions, assisting with dietary needs, replacing diapers, bathing, and providing mobility assistance in nursing hospitals, contribute to a heightened perception of fatigue [15]. Caregivers also reported rapid fatigue due to the need to suppress individual emotions and maintain constant emotional control for kindness and patience in interactions with patients, families, and medical staff [16]. Notably, factors such as sleep disorders and the demands of shift work in nursing hospitals were identified as contributors to increased fatigue [15,17]. When caregivers experience emotional exhaustion while performing their duties, it can lead to emotional states that pose a threat to mental health, such as anxiety and depression, ultimately resulting in a decline in the quality of life [18]. Medical practitioners generally experience reduced service quality as job stress increases [8]. Caregivers in nursing hospitals experience high physical fatigue and suffer from reduced sleep quality and changes of mood, such as depression due to job stress [17,19]. The job stress experienced by caregivers during duty can erode their confidence and self-efficacy, negatively affecting physical and mental health, as well as job performance [20]. This can lead to burnout [21], deteriorating the quality of care services and their quality of life [1].

Meanwhile, self-efficacy is a concept related to the evaluation of subjective cognitive abilities to successfully perform tasks and serves as a key factor in human achievement [22]. In other words, an individual’s belief in themselves influences their behavior, and the level of their behavior varies depending on how well they perceive their ability to perform the task [18]. Therefore, the self-efficacy of caregivers lowers job stress. In the relationship between the working environment and job stress, self-efficacy has a mediating effect, which helps reduce caregivers’ job stress and improve the quality of life [11,23,24]. In addition, humans grow and develop within various interpersonal relationships, with these connections serving as a key factor in determining the quality of human life [25]. Workplace relationships, in particular, significantly impact job satisfaction and overall quality of life [26]. Positive human relationships of female workers enhance their self-identity, increase their sense of stability, and promote good adaptation, thereby reducing stress, enhancing health, and improving the quality of life [27,28]. In the work environment, a harmonious and positive human relationship with colleagues and supervisors can reduce emotional exhaustion levels and have a mediating effect [28], thereby lowering caregiver burnout and ultimately improving the quality of life.

Therefore, the aim of this researcher is to comprehend the relationship among fatigue, depression, job stress, self-efficacy, interpersonal relationships, and the quality of life of nursing caregivers in nursing hospitals. In addition, through hierarchical analysis, the factors affecting the quality of life of care providers are closely analyzed, including general characteristics and fatigue, depression, job stress, self-efficacy, and interpersonal relationships. This approach helps identify changes in explanatory power, serving as foundational data for intervention programs aimed at enhancing the overall quality of life.

## 2. Conceptual Framework

The health-related quality of life model by Ferrans et al. [10] was referred to. This model distinctly elucidates the causal relationship between health-related quality of life and its associated elements. Not only physical health but also broader factors of life are mentioned, which aids in defining the scope of health-related quality of life. Factors that influence the quality of life such as biological function, symptoms, functional status, and general health perception were considered. It was explained that personal and environmental characteristics influence the quality of life through these factors.

Biological function pertains to the dynamic processes that sustain life, while symptoms include the recognition of various signs indicating abnormal physical, emotional, and cognitive states. Functional status refers to the capacity to perform physical, psychological, and social functions and roles, and general health perception denotes the subjective evaluation of overall health. Moreover, personal characteristics encompass developmental and psychological factors that influence health outcomes, and environmental characteristics include social and physical traits, encompassing inter-individual or social influences. Therefore, based on the model of Ferrans et al. [10], the researchers considered factors that are related to and presumed to influence caregivers’ quality of life. Fatigue was included as an overall health response that caregivers personally feel, under elements like biological function, functional status, and general health perception. Depression, which frequently manifests in women’s characteristics, was incorporated under symptoms. Self-efficacy, anticipated to have a mediating effect on fatigue and depression, was considered under personal traits, and job-related stress and human relationships, experienced by caregivers in nursing hospitals, were included under environmental characteristics. Thus, the intention was to discern the influence of these elements on the quality of life.

## 3. Materials and Methods

### 3.1. Participants

Participants included middle-aged women (41 years and older) and elderly women (65 years and older) who served as caregivers in nursing hospitals. There are about 40,000 caregivers working in nursing hospitals nationwide. As for the sample size, 137 female caregivers were conveniently gathered from 9 nursing hospitals in 6 cities located in the central region of Korea. The selection criteria are as follows:(1)Adult woman aged 19 or older according to the standards of civil law(2)Those working as personal care aides or nursing assistants in nursing hospitals.(3)Those with more than three months of caregiving experience for vulnerable elderly patients in nursing hospitals.(4)Those who signed the written consent for the study and expressed willingness to participate.(5)Individuals who are fluent in Korean and can comprehend and complete the questionnaire.

Based on a previous study [29] that used hierarchical regression analysis and the G-power 3.1.9.7. program for statistical analysis of multiple regression, with 13 predictor variables, an effect size of 0.15, a significance level of 0.05, and a power of 0.80, the required sample size was found to be 131. Considering a 5% dropout rate, 137 participants were surveyed, and all were included in the study.

### 3.2. Procedures

After obtaining approval from the Institutional Review Board (IRB) of the university, as for the sample size, 137 female caregivers were conveniently extracted from 9 nursing hospitals in 6 cities located in the central region of Korea. They explained the purpose and methods of the research to the directors and sought permission for data collection. A meeting was held with the nursing director or head nurse of each institution to explain the study and its procedures and obtain their cooperation. The questionnaires were administered directly by researchers to caregivers, explaining the study’s purpose and how to fill out the form. Consent was sought for those willing to participate. For participants not met directly, questionnaires were delivered through the nursing director or head nurse. The time taken to complete the survey was approximately 15–20 min. As a token of appreciation, participants were given a small gift.

Data collection was collected after distributing a paper questionnaire to the subject, coded in Excel to the computer, and called to SPSS Statistics 27.0 program for statistical analysis. After that, the coded data is stored in a file with a password on the computer owned by the lead researcher in accordance with the principles of the Institutional Bioethics Committee, making it difficult for others to access it. And keep the paper questionnaire in a bookshelf with a lock. After writing the paper, the file will be stored on the computer and the paper questionnaire will be stored on the bookshelf for three years, and the file will be deleted in a way that cannot be recovered and the paper questionnaire will be crushed with a shredder.

### 3.3. Measures

The research instrument consists of a questionnaire on general characteristics (11 items), fatigue (30 items), depression (9 items), self-efficacy (8 items), job stress (11 items), interpersonal relationships (7 items), and quality of life (26 items).

#### 3.3.1. Fatigue

The fatigue measurement tool used is the Subjective Symptoms of Fatigue Test proposed by the Japan Industrial Hygiene Association [30] in 1967 and finalized in 1970, translated into Korean by Yang and Han [31]. It consists of 30 items—10 physical symptoms, 10 mental symptoms, and 10 neurological symptoms. Each item is rated on a Likert scale from 1 (not at all) to 5 (very much), with higher scores indicating greater fatigue. The tool’s reliability at the time of development was Cronbach’s α = 0.91, while in Yang and Han’s study [31], it was α = 0.91. In this study, the overall reliability was α = 0.97, with subdomains ranging from 0.92–0.96.

#### 3.3.2. Depression

The depression measurement tool used is a modified version of the CES-D (the Center for Epidemiologic Studies—Depression Scale) [32] by Shin [33], which consists of 9 items after removing two with low reliability and validity. Each item is scored on a Likert scale from 0 (rarely, less than one day a week) to 3 (most of the time, more than six days a week). The questionnaire asks about depressive perceptions based on the psychological state of the past week, with higher scores indicating greater depression. In Shin’s study [33], the reliability was Cronbach’s α = 0.87, while in this study, it was α = 0.91.

#### 3.3.3. Job Stress

The job stress tool used Jayaratne’s tool [34], which was adapted and employed by Shin [35]. This adaptation utilized the scale created by Lee and Park [36] for domestic circumstances, with the scale developed by Dietary and Schneider [37] serving as a reference. Researchers deleted three items with low reliability and validity. Finally, the tool consists of 11 items. Each item is rated on a Likert scale from 1 (not at all) to 5 (very much), with higher scores indicating greater job stress. In Lee & Park’s study [36], the reliability by half reliability was 0.85, and in this study, it was α = 0.88, with subdomains ranging from 0.68–0.82.

#### 3.3.4. Self-Efficacy

The self-efficacy tool was developed by Chen, Gully, & Eden [38] in 2001. It consists of 8 items rated on a 5-point Likert scale, with higher scores indicating higher self-efficacy. In Chen et al.’s study [38], the reliability was Cronbach’s α = 0.87, and in this study, it was α = 0.93.

#### 3.3.5. Interpersonal Relationship

The interpersonal relationship tool was adapted from the scale by Soderfeldt [39] and was employed by Kong [28]. It comprises 7 items based on a 5-point Likert scale. Scores range from “not at all” (1 point) to “very much so” (5 points), with higher scores indicating better interpersonal relationships. In the study by Kong [28], the reliability was Cronbach’s α = 0.85, while it was 0.89 in this research.

#### 3.3.6. Quality of Life

The quality of life was measured using the Korean version of the WHOQOL-BREF developed by the WHOQOL Group [40] and translated by Min et al. [41]. The tool consists of a total of 26 items categorized into 5 domains: 2 items on overall quality of life and general health, 7 items on physical health, 6 items on psychological health, 3 items on social relationships, and 8 items on the living environment. Each item is based on a 5-point Likert scale ranging from “not at all” (1 point) to “very much so” (5 points). Scores for the 3 reverse-scored items were inverted, with higher scores denoting better quality of life. Scoring was based on the manual, converting the total score for all items and scores for each domain to a maximum of 100 points. In the study by Min et al. [41], the reliability was Cronbach’s α = 0.92, while it was 0.90 in this research, and the reliability for the sub-domains ranged from 0.71 to 0.83.

### 3.4. Data Analysis

The SPSS Statistics 27.0 program (IBM Corporation, Armonk, NY, USA) was used for the analysis. Technical statistics, such as real numbers and percentages, were employed to describe the general characteristics and variables of the study subjects. Differences in the QoL based on the general characteristics of the study subjects were assessed using *t*-tests and ANOVA, with post-tests analyzed using the Scheffe test. The correlation between fatigue, depression, self-efficacy, job stress, interpersonal relationships, and the QoL of the subjects was analyzed using Pearson’s correlation coefficients.

As a method of analyzing factors affecting the QoL of care providers, first, the influencing factors among the general characteristics of the subject were identified, followed by factors such as biological function, functional status, and general health perception as the overall health response of care providers, and then environmental factors and variables that can provide mediating effects were analyzed. Therefore, a hierarchical analysis was performed to analyze the results and explanatory power of factors affecting quality of life through this hierarchical analysis.

### 3.5. Ethical Consideration

The study was approved by the Institutional Review Board of K National University (KNU_IRB_2023-92) and ensured compliance with ethical guidelines regarding the purpose, methods, and protection of participants’ rights. The consent form provided participants with information about anonymity and confidentiality. Participants were informed that they could withdraw from this study at any time and would not face any penalties. Personal information collected was managed in accordance with privacy laws, and the researchers made every effort to maintain the confidentiality of all information obtained. Informed consent was obtained from all participants, and the data collected were stored securely in a locked bookcase accessible only to the researchers for three years. After the study was completed, participants were informed that research-related materials would be retained for three years and then securely destroyed using a shredder.

## 4. Results

### 4.1. General Characteristics of Participants

The general characteristics of the study participants are presented in Table 1. The age of the participants ranged from 52 to 76 years, with an average age of 65.85 ± 4.44 years. The majority, 98 participants (71.5%), were between 60 and 69 years of age. As for religion, 75 participants (54.7%) reported having no religious affiliation. The majority were married, with 122 participants (89.1%), and 103 of them (75.2%) were satisfied with their marriage. Regarding educational attainment, the majority, 130 participants (94.8%), had an education level of high school or below. Fifty two participants (38.0%) had been working as caregivers for less than 5 years. Eighty-three participants (60.6%) had no experience receiving education related to the quality of life. Seventy-two participants (52.6%) reported a monthly income of over 2 million won, while 94 participants (68.6%) assessed their economic status as average. A significant majority, 108 participants (78.8%), did not engage in hobbies or leisure activities. Furthermore, 104 participants (75.9%) had one or more diseases.

### 4.2. Differences in QoL According to General Characteristics of Participants

The differences in quality of life based on the general characteristics of the participants were as follows (Table 1). There were statistically significant differences in the degree of QoL associated with age (F = 3.91, *p* = 0.022), marital status (t = 2.65, *p* = 0.009), marital satisfaction (t = −4.05, *p* < 0.001), education level (t = −2.06, *p* = 0.041), experience of QoL education (t = −2.20, *p* = 0.029), monthly income (t = −2.61, *p* = 0.010), perceived economic status (t = 7.18, *p* = 0.001), engagement in hobbies or leisure activities (t = −3.02, *p* = 0.003), and number of illnesses (t = −2.97, *p* = 0.004). Specifically, participants aged 70 and above had a higher quality of life compared to those aged 60–69. Those who were married had a higher QoL compared to those who were single, divorced, or widowed. Higher marital satisfaction, having tertiary education compared to high school or below, receiving QoL education more than once, having a monthly income of more than 1500 dollars, evaluating one’s economic status as average rather than poor, having hobbies or leisure activities, and being without any diseases all correlated with a higher QoL. No significant differences were observed in relation to religion, work experience in nursing hospitals, or experience as a caregiver.

### 4.3. Degree of Fatigue, Depression, Job Stress, Self-Efficacy, Interpersonal Relationship and QoL of Participants

The degrees of fatigue, depression, self-efficacy, job stress, interpersonal relationships, and quality of life among the participants were as follows (Table 2). The participants’ fatigue scored an average of 2.17 ± 0.70 out of 5, with physical symptoms slightly higher at 2.43 ± 0.73, and neurological symptoms the lowest at 1.96 ± 0.81. Depression was scored at 0.84 ± 0.56 out of 3. Job stress was scored at 3.00 ± 0.59 out of 5, with role ambiguity being the highest at 3.11 ± 0.76 and role overload the lowest at 2.97 ± 0.59. Self-efficacy scored at 3.39 ± 0.70 out of 5. Interpersonal relationships scored 3.32 ± 0.60 out of 5. Quality of life averaged 63.8 ± 8.00 out of 100, with psychological health being the highest at 66.0 ± 9.48 and the social relationship domain the lowest at 61.8 ± 9.83.

### 4.4. Correlations between Fatigue, Depression, Job Stress, Self-Efficacy, Interpersonal Relationship and QoL of Participants

The correlations between the participants’ fatigue, depression, self-efficacy, job stress, interpersonal relationships, and quality of life are presented as follows (Table 3). The quality of life of the participants showed statistically significant correlations with fatigue (r = −0.30, *p* < 0.001), depression (r = −0.31, *p* < 0.001), job stress (r = −0.17, *p* = 0.043), self-efficacy (r = 0.51, *p* < 0.001), and interpersonal relationships (r = 0.67, *p* < 0.001). In other words, a lower level of fatigue, lower depression, less job stress, higher self-efficacy, and better interpersonal relationships were associated with a higher quality of life among the participants.

### 4.5. Factors Affecting QoL of Participants

The analysis results regarding the factors affecting the QoL of the participants are as follows (Table 4). Initially, to test the effects of participants’ fatigue, depression, self-efficacy, job stress, and interpersonal relationships on QoL, the basic assumptions of multiple regression analysis were verified. As a result of the hypothesis test, a value of 1.48, close to 2, was derived in the Durbin-Watson test, indicating the absence of autocorrelation and independence among the model’s error terms. The normal P-P plot showed residuals in a straight line, indicating a normal distribution. The range of tolerance limits was from 0.44 to 0.83, and the VIF value range was from 1.21 to 2.28, indicating no multicollinearity issues.

In the first step of the hierarchical regression analysis, variables that showed differences in QoL based on the general characteristics of the participants, such as age, marital status, marital satisfaction, education, educational experience for improving QoL, monthly income, perceived economic status, and hobbies or leisure activities, were included in the analysis as a control variable. These variables underwent dummy coding. The results showed that perceived economic status (β = −0.18, *p* = 0.033) and hobbies or leisure activities (β = 0.19, *p* = 0.025) had a significant impact on QoL, explaining 26.6% of the variance (F = 4.58, *p* < 0.001).

In the second step, depression and fatigue, which are related to QoL, were included, and an additional 4.0% explanatory power was added. Analysis results showed that monthly income (β = −0.19, *p* = 0.034) and perceived economic status (β = −0.18, *p* = 0.035) were significant predictors of QoL, explaining 30.6% (F = 4.56, *p* < 0.001).

In the third step, potential mediators like self-efficacy, job stress, and interpersonal relationships were added, and an additional explanatory power of 30.9% was gained. The analysis results indicated that age (β = −2.80, *p* = 0.006), perceived economic status (β = −2.41, *p* = 0.017), self-efficacy (β = 3.19, *p* = 0.002), and interpersonal relationships (β = 7.12, *p* <0.001) were significant predictors of QoL, accounting for a total of 61.5% (F = 12.88, *p* < 0.001). Interpersonal relationships were the strongest predictor affecting participants’ QoL.

## 5. Discussion

This study aimed to determine the impact of fatigue, depression, job stress, self-efficacy and interpersonal relationships on the quality of life (QoL) of middle-aged and elderly women providing care for the elderly and patients with chronic diseases in nursing hospitals. The study hierarchically regressed these factors focusing on general characteristics, fatigue, depression, job stress, self-efficacy, and interpersonal relationships.

Initially, there were differences in QoL based on the participants’ age, marital status, marital satisfaction, education, educational experience regarding QoL, monthly income, perceived economic status, hobbies or leisure activities, and the number of illnesses. This supports previous findings that adult female workers’ QoL varied according to age, education, monthly income, and marital status [42]. For married working women, QoL varied based on age, education, monthly income, and leisure activities [43]. Middle-aged working women’s QoL differed depending on their education and perceived economic status [15]. Additionally, the well-being of caregivers varied based on their educational experience on well-being and monthly income [29]. These findings align with and support the results of this study.

However, findings related to age differed from previous research. Caregivers in Korean nursing hospitals are typically women in their middle age or later, differing from the general age characteristics of female workers. Analyzing the higher QoL of caregivers aged 70 and above compared to those in their 50 s and 60 s, it can be inferred that those over 70 have longer work and caregiving experience, leading to higher job proficiency. Such proficiency might reduce job-related stress and increase satisfaction in caring for the elderly and chronically ill patients, thereby enhancing their perceived QoL. Therefore, it is suggested that future studies consider this aspect and repeatedly research the QoL of caregivers based on their age. Moreover, administrators and decision-makers in nursing hospitals need to consider factors such as age, marital status, marital satisfaction, education, educational experience about QoL, perceived economic status, hobbies or leisure activities, and the number of illnesses when devising policies and systems for caregivers’ welfare.

In terms of education and perceived economic status, individuals with a college degree or higher experienced a higher quality of life compared to those with only a high school diploma. Similarly, those with a monthly income of $1500 or more enjoyed a higher quality of life than those earning less than $1500. Furthermore, the group that received higher education demonstrated a higher quality of life than the group that did not, and those engaged in hobbies and leisure activities reported a higher quality of life than those who were not involved in such activities. These findings underscore the correlation between education, economic factors, and overall quality of life. The higher quality of life among individuals with advanced education is attributed to their elevated economic achievement and the diverse opportunities for life activities available to them [1,15].

Monthly income and economic status play a pivotal role in enabling individuals to engage in hobbies and leisure activities, aligning with the findings that those with such pursuits experience a higher quality of life compared to those without [43]. Therefore, to enhance the quality of life for care providers in nursing hospitals and foster satisfaction in their work, it is imperative for the individuals overseeing these institutions to develop educational programs and ensure continuous education. Additionally, recognizing the significance of economic aspects, it is crucial to provide caregivers with appropriate compensation. Measures should be devised to establish clubs for hobbies and leisure activities, along with allowances to promote the overall welfare of the employees.

In addition, individuals who are married or report high marital satisfaction exhibit a higher quality of life compared to those experiencing bereavement, divorce, or dissatisfaction with marriage. This quality of life is also higher than that observed in disease-free cases. The marital satisfaction of middle-aged women has been found to be positively correlated with their quality of life, contributing to psychological stability [44]. Marriage status and marital satisfaction serve as sources of social support, encompassing assistance from both spouses and family. Spousal support emerges as a crucial variable for maintaining psychological stability and overall health [44]. Similarly, family consideration and assistance constitute support systems that aid caregivers in effectively managing their roles in the workplace [1]. In instances of unsatisfactory marriages, emotional crises, such as helplessness, despair, and unhappiness, are commonly experienced. Thus, there is a need to formulate strategies aimed at increasing social support to enhance the quality of life for nursing care workers.

Furthermore, it aligns with reported findings that the presence of a disease leads to decreased work efficiency, deterioration in health, prolonged return-to-work periods, and ultimately lower quality of life for workers [42]. Consequently, it is essential to make concerted efforts to maintain and promote the health of caregivers themselves. This includes focusing on disease prevention, regular health check-ups, and the steadfast maintenance of healthy lifestyles in their daily lives.

Regarding the levels of fatigue, depression, self-efficacy, job stress, human relationships, and quality of life of the participants, their fatigue was low at 2.17 points, but it was somewhat high in the physical symptom domain. Depression was also low at 0.84 points out of a 3-point scale. Self-efficacy was moderate at 3.39 points, and job stress was also moderate at 3.00 points. Among the sub-domains, role ambiguity was the highest at 3.11 points but was still considered moderate. Human relationships were moderate at 3.32 points out of 5, and the quality of life was also moderate at 63.8 points, with the social relationship sub-domain being the lowest at 61.8 points, yet still considered moderate. The fatigue of middle-aged caregivers in charge of home visits was 2.04 out of 5 points, and job stress was also slightly low at 2.45 points [45]. This was consistent with this study. Moreover, the self-efficacy of middle-aged caregivers was 3.80 points [46], the human relationships of caregivers in elderly care facilities were 3.62 out of 5 points [28], and the quality of life of middle-aged working women was 71.05 points [15]. These findings were consistent with and supported the results of this study. In many studies targeting female caregivers after middle age, the participants had low levels of fatigue, depression, and job stress, and their self-efficacy and human relationship levels were moderate. Therefore, workers and managers of nursing hospitals should strive to build beliefs about their jobs, foster harmonious social human relationships, and create work environments and welfare systems that they desire.

In the correlation results between the participants’ quality of life and related variables, the higher the quality of life of the participants appeared as their fatigue was lower, depression was lower, job stress was lower, self-efficacy was higher, and human relationships were better. Furthermore, when analyzing factors affecting the quality of life of the participants, perceived economic status and hobbies or leisure activities had a significant influence on quality of life in the first stage. Monthly income and perceived economic status were significant predictors of quality of life in the second stage. In the third stage, age, perceived economic status, self-efficacy, and interpersonal relationships influenced, explaining a total of 61.5% (Figure 1).

The QoL of caregivers is influenced by social, physical, and psychological aspects [47,48]. Fatigue is an empirical phenomenon that leads to a decrease in the ability to perform daily activities when the burden increases due to excessive mental and physical exertion. It serves as a sign of homeostasis disruption and contributes to a decline in physical, mental, and emotional capabilities. Studies have reported that stress and depression are risk factors associated with this fatigue [49]. Middle-aged working women, in particular, experience higher levels of job stress compared to their counterparts [50]. This heightened stress, coupled with physical and psychological burdens, results in fatigue, ultimately leading to a diminished quality of life [15,51]. In addition, Kwon’s results [52] indicate that the quality of life of nurses working in hospitals is inversely proportional to the levels of depression and job stress. In particular, middle-aged women over 41 who simultaneously manage work and household chores were found to experience increased fatigue and stress [27]. This is particularly true for shift workers, as fatigue persists even after sleep. If not addressed, persistent fatigue can pose serious threats to physical and mental health, leading to issues such as overwork, reduced productivity, cardiovascular problems, cancer, and depression [53]. Given these challenges, for caregivers compelled to work shifts due to the nature of their job, it is imperative to implement measures aimed at reducing fatigue, job stress, and depression.

On the other hand, although there exists a correlation between fatigue, depression, job stress, and quality of life, the study did not find these factors to be direct determinants of caregivers’ quality of life. The quality of life for caregivers is influenced by various factors, including the severity of the disease in dementia patients admitted to nursing homes, the caregiver’s perception of dementia, and family of the patients’ care needs [54]. A higher number of employees correlates with a higher quality of life for caregivers [55,56], suggesting a positive impact on the overall quality of service. In essence, the number of caregivers, the severity of dementia in patients, and family of the patients’ care needs collectively affect the caregiver’s quality of life. These factors can serve as significant prerequisites for caregivers to experience fatigue, depression, and job stress in the performance of their duties. Surprisingly, in this study, fatigue, depression, and job stress did not emerge as major influencing factors, presenting results that differ from previous studies. Consequently, it is imperative to conduct further studies, taking into account these aforementioned factors in future research. Such investigations are crucial for a comprehensive understanding of the dynamics affecting caregiver well-being.

Furthermore, this study identified self-efficacy as a major factor in enhancing the quality of life. It was found that the higher the self-efficacy of caregivers, the lower the job stress [11] and the higher the service quality in long-term care institutions when caregivers have high self-efficacy and low job stress, confirming the significant influence of self-efficacy on service quality [12]. This finding aligns with results that suggest higher self-efficacy leads to job satisfaction and an improved quality of life [57], further supporting our research. In a study targeting caregivers, it was found that self-regulation efficacy, a sub-domain of self-efficacy, significantly impacts the quality of life [58]. When self-regulation efficacy is high, individual achievements increase, thus positively influencing the quality of life [59]. Hence, consideration of self-regulation efficacy is essential.

Individuals with high self-efficacy adapt well to their environment, remaining calm even in stressful situations and effectively coping with unsatisfactory circumstances [60]. Those with low self-efficacy might overestimate their inadequacies in stressful situations, inhibiting them from efficiently utilizing their capabilities [61]. Therefore, strategies to boost self-efficacy, ensuring effective coping, are needed. Self-efficacy refers to the confidence that one can effectively cope in any given situation, expressed as ‘I am capable’, ‘I can overcome’, and ‘I can achieve my goals’ [11]. Such self-efficacy has a crucial impact on an individual’s quality of life [62]. As the quality of life increases with higher self-efficacy and enhances the sense of accomplishment, strategies to enhance self-efficacy are essential.

The results of this study revealed that the most significant factor influencing the quality of life of caregivers was interpersonal relationships. For Koreans, interpersonal relationships form the foundation of emotions and hold an absolute importance in life. In traditional Korean culture, the basic axis of social relations is the ‘ourist group’.In other words, in the social relationship of Koreans, an individual is not independent, but rather a ‘relationship individual’ who becomes one with another called ‘we’. Koreans establish an interpersonal relationship framework within our category, focusing on school ties, regional ties, and blood ties, believe that it is a safe relationship, thoroughly preserve and strengthen the relationship, unconditionally accept it, and have full trust [63]. Hence, they are of paramount significance and much time and effort are invested to establish, maintain, and enhance them. Therefore, interpersonal relationships are a key factor determining the quality of life for Koreans [26,62,64]. Other factors included economic strength, health, and leisure activities. Human relationships play a role as the linkage in the social structure [49]. Additionally, relationships within the workplace serve as the source of productivity and determine the nature and content of one’s work life. Notably, the significant factor affecting the job satisfaction of caregivers working in elderly residential welfare facilities and medical welfare facilities is interpersonal relationships [65]. This indicates that when caregivers, in a hospital setting, maintain harmonious relationships with other healthcare professionals and perform their duties, their satisfaction and quality of life can improve. There are both formal relationships due to work and informal personal relationships based on daily interactions, emotions, or value judgments [66]. Efforts should be made to ensure that both formal and informal relationships are well-established to enhance the quality of life. For maintaining such relationships successfully, social relations or networks are essential. The more connections in a network, the more active the communication, leading to similarities in attitudes, opinions, and behaviors among members. This facilitates access to information and conversations and results in higher satisfaction from relationships with others [66]. Thus, the development and application of relationship network mediation programs within the institution, both inter-departmental and intra-departmental, where the caregiver is affiliated, are required.

As a limitation, care should be taken when interpreting the results because this study focused on specific areas in the central region. In this study, only those who could communicate in Korean were selected as subjects. Therefore, it is suggested that the research be expanded in the future, including foreign caregivers working in nursing hospitals. At this time, we need the help of a foreign leader who can interpret.

## 6. Conclusions

The purpose of this study was to identify the factors affecting the quality of life of middle-aged women over 41 and elderly women over 65 serving as caregivers in nursing hospitals. In the first phase, general characteristics such as perceived economic status and hobbies or leisure activities influenced the quality of life. In the second phase, even when factors like depression and fatigue were included in the analysis, monthly income and economic status still affected the quality of life. In the third phase, apart from age and economic status, self-efficacy and interpersonal relationships were revealed as mediating factors influencing the quality of life. Currently, caregivers in nursing hospitals, such as nursing aides or private nurses, who provide care to critically ill elderly or vulnerable patients with chronic diseases, emphasize that their quality of life is of utmost importance in order to offer superior care services. Therefore, based on the findings of this study, there is an immediate need to develop and implement intervention programs that can enhance self-efficacy and interpersonal relationships. This research will serve as foundational data to guide both quantitative and qualitative studies aimed at improving the quality of life of the participants. Furthermore, leaders in nursing hospitals need to make provisions for policies and systems that cater to raising monthly salaries to improve perceived economic conditions, establish clubs for hobbies and leisure activities to enhance caregiver welfare, and offer legislative support.

It was found that variables such as fatigue, depression, and job stress, which were theoretically likely to be related, did not affect. Therefore, repeated studies are needed for caregivers working in nursing hospitals in Korea. Continuous research is also needed to identify other factors in order to increase the explanatory power of factors affecting quality of life. Considering the variables of this study, we propose a structural model study that can grasp direct and indirect effects and total effects. Self-efficacy and interpersonal relationships are crucial factors for enhancing the quality of life for middle-aged women over 41 and elderly women over 65 serving as caregivers in nursing hospitals. Therefore, specific research and intervention development on self-efficacy and interpersonal relations are needed.

## Figures and Tables

**Figure 1 behavsci-14-00053-f001:**
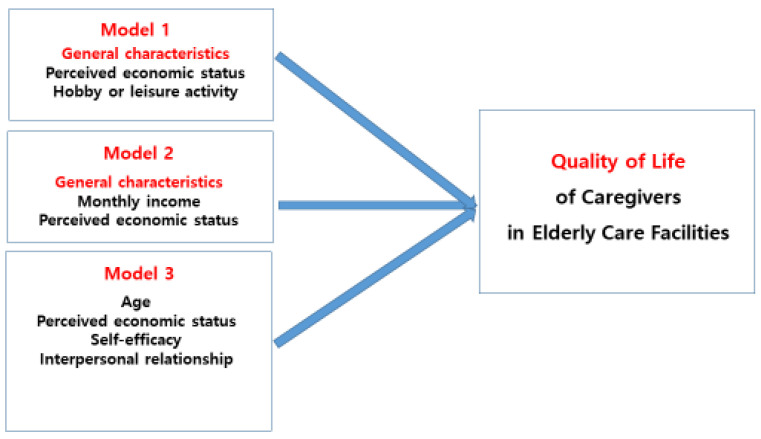
Factors influencing QoL of caregivers.

**Table 1 behavsci-14-00053-t001:** General characteristics of the participants and the differences in QoL according to general characteristics. (N = 137).

Variables	Classification	*n*	%	QoL		t/F	*p*-Value Scheffe Test
				Mean	SD		
Age (years)	52–59	11	8.0	3.21	0.18	3.91	0.022
	60–69	98	71.5	3.13	0.39		c > b
	≥70	28	20.4	3.37	0.46		
Religion	Have	62	45.3	3.24	0.42	1.41	0.180
	Don’t have	75	54.7	3.14	0.38		
Marriage	Married	122	89.1	3.22	0.40	2.65	0.009
	Widowed, divorced, or separated	15	10.9	2.93	0.31		
Marital satisfaction	Unsatisfied	34	24.8	2.96	0.31	−4.05	<0.001
	Satisfied	103	75.2	3.26	0.40		
Education	Under graduated from high school or lower	130	94.8	3.17	0.40	−2.06	0.041
	Graduated from college or higher	7	5.2	3.49	0.35		
Work experience as a caregiver	<5	52	38.0	3.17	0.31	1.63	0.200
(years)	5–<10	41	29.9	3.12	0.46		
	≥10	44	32.1	3.27	0.44		
Educational experience of QoL	None	83	60.6	3.13	0.42	−2.20	0.029
{number}	≥1	54	39.4	3.28	0.36		
Monthly income	<1500	65	47.4	3.10	0.36	−2.61	0.010
(dollars)	≥1500	72	52.6	3.28	0.42		
Perceived economic status	High	3	2.2	3.04	0.00	7.18	0.001
	Medium	94	68.6	3.27	0.39		b > c
	Law	40	29.2	3.00	0.37		
Hobby or leisure activities	Don’t have	108	78.8	3.14	0.37	−3.02	0.003
	Have	29	21.2	3.38	0.42		
Number of disease	0	33	24.1	3.36	0.48	2.97	0.004
(number)	≥1	104	75.9	3.13	0.36		

SD = standard deviation. QoL = quality of life.

**Table 2 behavsci-14-00053-t002:** Degree of fatigue, depression, job stress, self-efficacy, interpersonal relationship and QoL of participants (N = 137).

Variables	Mean	Meaning	SD	Actual Range
Fatigue	2.17	Mild	0.70	1.00–3.93
Physical symptoms	2.43	Mild-moderate	0.73	1–4
Psychological symptoms	2.14	Mild	0.77	1–4
Neurosensory symptoms	1.96	Mild	0.81	1–3.80
Depression	0.84	Mild	0.56	0–2.22
Job stress	3.00	Moderate	0.59	1–4.55
Role conflict	2.99	Moderate	0.75	1–4.5
Role ambiguity	3.11	Moderate	0.76	1–4
Be too much of a role	2.97	Moderate	0.59	1–5
Self-efficacy	3.39	Moderate	0.70	1–5
Interpersonal relationship	3.32	Moderate	0.60	1–5
QoL	63.8	Moderate	8.00	44.6–93.0
Overall QoL, general health	63.2	Moderate	13.60	20–100
Physical health	62.8	Moderate	9.49	40–94.20
Psychological health	66.0	Moderate	9.48	40–96.60
Social relationship	61.8	Moderate	9.83	33.40–86.60
Environment	63.6	Moderate	9.45	35–97.60

SD = standard deviation, QoL = quality of life.

**Table 3 behavsci-14-00053-t003:** Relationships between of fatigue, depression, self-efficacy, job stress, interpersonal relationship and QoL of participants (N = 137).

Variables	Fatigue r (*p*)	Depression r (*p*)	Job Stress r (*p*)	Self-Efficacy r (*p*)	Interpersonal Relationship r (*p*)	Quality of Life r (*p*)
Fatigue	1					
Depression	0.66 (<0.001)	1				
Job stress	0.23 (0.007)	0.29 (0.001)	1			
Self-efficacy	−0.19 (0.025)	−0.23 (0.006)	0.16(0.064)	1		
Interpersonal relationship	−0.24 (0.005)	−0.38 (<0.001)	−0.22 (0.009)	0.47 (<0.001)	1	
Quality of life	−0.30 (<0.001)	−0.31 (<0.001)	−0.17 (0.043)	0.51 (<0.001)	0.67 (<0.001)	1

**Table 4 behavsci-14-00053-t004:** Factors affecting the QoLof participants.

Variables	Model 1					Model 2					Model 3				
	B	SE	β	t	*p*	B	SE	β	t	*p*	B	SE	β	t	*p*
Constants	3.15	0.11		28.79	<0.001	3.68	0.21		17.44	<0.001	1.85	0.24		7.65	<0.001
Age (52–59)	−0.03	0.12	−0.02	−0.26	0.797	−0.06	0.14	−0.04	−0.48	0.635	−1.67	0.10	−0.11	−1.62	0.108
Age (≥70)	0.16	0.08	0.16	1.91	0.059	−0.12	0.08	−0.13	−1.41	0.160	−0.18	0.06	−0.20	−2.80	0.006
Marriage (widowed, divorced, or separated)	−0.20	0.10	−0.15	−1.94	0.054	−0.16	0.10	−0.13	−1.63	0.106	−0.01	0.08	−0.01	−0.04	0.969
Marital satisfaction (satisfied)	0.12	0.09	0.13	1.42	0.158	0.09	0.09	0.10	1.08	0.284	0.06	0.07	0.06	0.90	0.369
Education (college or higher)	0.11	0.15	0.06	0.77	0.440	−0.11	0.14	−0.06	−0.74	0.464	0.06	0.11	0.04	0.58	0.565
Educational experience of QoL (≥1)	0.02	0.07	0.03	0.30	0.763	0.03	0.07	0.03	0.40	0.694	0.01	0.05	0.01	0.18	0.856
Monthly income (≥1500 dollars)	−0.13	0.07	−0.16	−1.78	0.077	−0.16	0.07	−0.19	−2.15	0.034	−0.05	0.06	−0.06	−0.83	0.409
Perceived economic status (high)	−0.02	0.23	−0.01	−0.10	0.918	−0.03	0.22	−0.01	−0.13	0.898	−0.08	0.17	−0.03	−0.45	0.656
Perceived economic status (low)	−0.16	0.08	−0.18	−2.16	0.033	−0.16	0.07	−0.18	−2.14	0.035	−0.14	0.06	−0.15	−2.41	0.017
Hobby or leisure activity	0.18	0.06	0.19	2.27	0.025	0.10	0.09	0.11	1.23	0.223	0.11	0.06	0.11	1.67	0.098
Fatigue						−0.09	0.06	−0.16	−1.52	0.131	−0.08	0.05	−0.15	−1.80	0.075
Depression						−0.07	0.07	−0.10	−0.95	0.344	0.10	0.56	0.14	1.75	0.082
Job stress											−0.01	0.05	−0.01	−0.17	0.869
Self-efficacy											0.13	0.04	0.23	3.19	0.002
Interpersonal relationship											0.34	0.05	0.51	7.12	<0.001
R^2^	0.266					0.306					0.615				
Adjusted R^2^	0.208					0.239					0.567				
Δ Adjusted R^2^ (*p*)						0.04 (<0.001)					0.309 (<0.001)				
F (*p*)	4.58 (<0.001)					4.56 (<0.001)					12.88 (<0.001)				

SE = standard error, QoL = quality of life; reference; age (52–59 = 0, 60–69 = 1), marriage (married = 0, widowed, divorced, or separated = 1), marital satisfaction (unsatisfied = 0, satisfied = 1), education (high school graduated or lower = 0, college graduated or higher = 1), educational experience of QoL (none = 0, ≥1 = 1), monthly income (<1500 dollars = 0, ≥1500 dollars = 1), perceived economic status (high = 0, low = 1), hobby or leisure activity(don’t have = 0, have = 1).

## Data Availability

The data underlying this article will be shared upon reasonable request from the corresponding author.

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
