# Peer review of "Determinants of Quality of Life (QoL) in Female Caregivers in Elderly Care Facilities in Korea"

_behavsci, 2024, doi:10.3390/bs14010053_

Round 1
Reviewer 1 Report (Previous Reviewer 3)
Comments and Suggestions for Authors
The authors made almost the suggestions I recommended in their first submission.
Comments on the Quality of English LanguageNothing to declare.
Author Response
|
Reviewer 1 |
Thank you for your sincere review opinion. I revised it hard to reflect your review. |
|
The authors made almost the suggestions I recommended in their first submission. |
I thoroughly understood and revised what you reviewed, and then revised it again to post a paper. As you pointed out, I revised the parts to be revised and earnestly presented my opinion on the part where I disagree. Thank you for your review to make it a good paper. |
Reviewer 2 Report (New Reviewer)
Comments and Suggestions for Authors
I consider that the topic addressed in the article is interesting and current, because everything that science can contribute to the care of the caregiver is important. Also, thinking that the population in the most developed countries is getting older every day. It is true that in recent decades the variables that can influence caregiver stress and burnout have been studied in some depth, but as I have indicated before, I believe it is essential to continue supporting the research and consolidating the results obtained previously. Therefore, I think that even if it is not very original, it is relevant within the field of study. In this case, the research supports other previously obtained results, ensuring knowledge in this area and with a specific population of a specific country. For this reason, I would include in the title the country where the study was carried out to make it very clear, since culture and society influence the care applied by the caregiver. The theoretical review is correct, as is the presentation of the results and their discussion. I also think that the conclusions raised are timely, perhaps the possible limitations of the study should be included or dealt with in greater depth.
Author Response
|
Reviewer 2 |
Thank you for your sincere review opinion. I revised it hard to reflect your review. |
|
1. In this case, the research supports other previously obtained results, ensuring knowledge in this area and with a specific population of a specific country. For this reason, I would include in the title the country where the study was carried out to make it very clear, since culture and society influence the care applied by the caregiver.
2.The theoretical review is correct, as is the presentation of the results and their discussion. I also think that the conclusions raised are timely, perhaps the possible limitations of the study should be included or dealt with in greater depth. |
Thank you very much for your deep understanding of this study. 1.The title includes Korea. Determinants of Quality of Life (QoL) in Female Caregivers in Elderly Care Facilities in Korea
->As a limitation, care should be taken when interpreting the results because this study focused on specific areas in the central region. In this study, only those who could communicate in Korean were selected as subjects. Therefore, research including foreign caregivers working in nursing hospitals and accompanied by interpretation is needed in the future. -> It was found that variables such as fatigue, depression, and job stress, which were theoretically likely to be related, did not affect. Therefore, repeated studies are needed for caregivers working in nursing hospitals in Korea. Continuous research is also needed to identify other factors in order to increase the explanatory power of factors affecting quality of life. Considering the variables of this study, we propose a structural model study that can grasp direct and indirect effects and total effects. Self-efficacy and human relationships are important influencing factors in order to improve the quality of life of middle-aged and elderly female caregivers. Therefore, specific research and intervention development on self-efficacy and human relations are needed.
|

Reviewer 3 Report (New Reviewer)
Comments and Suggestions for Authors
This study touches on a subject of much worry and research in the helping professions - fatigue and burnout among formal and informal caregivers of people who require special daily care and help in ADL, etc. The study is of added value and I think it can be of interest to the journals audiences, but it may require some additional tweaking before it is ready for publication:
1. English language: while generally the language used in the paper is flowing and clear some point require the attention of a native speaking English language editor.
2. In the introduction it is not very clear what the aim of the study is until the last paragraph. I would try to present the problem of QOL of employees in the helping p[rofessions in long term care facilities at a very early stage of the introduction and present the importance of understanding their QOL (e.g.: is there a high rate of turnover in this industry? lack of trained or high quality personnel? issues with quality of care that can be traced back to employees QOL?) Then present the background and the settings for the study as well as its rationale.
3. Try to provide a more comprehensive picture of global findings in this venue - it is a subject that is well studied around the world and can be reported and covered from various angles throughout the world, for example:
Winzelberg, G. S., Williams, C. S., Preisser, J. S., Zimmerman, S., & Sloane, P. D. (2005). Factors associated with nursing assistant quality-of-life ratings for residents with dementia in long-term care facilities. The Gerontologist, 45(suppl_1), 106-114.
Noelker, L. S., & Harel, Z. (2001). Humanizing long-term care: Forging a link between quality of care and quality of life. Linking quality of long-term care and quality of life, 3-26.
Bowers, B. J., Esmond, S., & Jacobson, N. (2000). The relationship between staffing and quality in long-term care facilities: Exploring the views of nurse aides. Journal of nursing care quality, 14(4), 55-64.
Shin, J. H., & Bae, S. H. (2012). Nurse staffing, quality of care, and quality of life in US nursing homes, 1996–2011: an integrative review. Journal of gerontological nursing, 38(12), 46-53.
And many more.
4. In the sample selection - the criteria for selection is unclear - why only women aged 41 and above, for example?
5. In the procedure there is no point in assigning code names (city A city B etc.) - either provide the actual city names or just say " a sample taken from 6 different cities in S. Korea" that can be enough.
6. Data analyses and results: I see little point in dividing variables values into categorical data. What is the meaning the answers yes or no to religion? which religion? what level of religiosity? same goes with age or work experience - leave the numbers (age in years, experience in years) as they are and gain more statistical power in your regression results!! If this is impossible for some reason at least explain why you chose to go this way.
7. I see no added value in a series of ANOVA or t-tests - step-wise regression in which demographics are entered in the first step, then the main variables are entered will provide the same data with more information about the nature of the effects found.
8. The discussion is decent. Please refrain from any language suggesting causation: speak of associations, explained variance but not influence since you utilized a correlational study design which does not allow the establishing of causal relationships between variables.
Develop the limitations section, refer not only to the sample, but also to the potential intervening nature of a specific culture, the use of specific questionnaires, etc.
Comments on the Quality of English Language
Should be edited for language by a native speaker of English.
Author Response
|
Reviewer 3 |
Thank you for your sincere review opinion. I revised it hard to reflect your review. |
|
1.English language: while generally the language used in the paper is flowing and clear some point require the attention of a native speaking English language editor. |
This is a paper I received after requesting translation from a famous translation company in Korea. I will review it again. Thank you. |
|
2.In the introduction it is not very clear what the aim of the study is until the last paragraph. I would try to present the problem of QOL of employees in the helping professions in long term care facilities at a very early stage of the introduction and present the importance of understanding their QOL (e.g.: is there a high rate of turnover in this industry? lack of trained or high quality personnel? issues with quality of care that can be traced back to employees QOL?) Then present the background and the settings for the study as well as its rationale. |
As you pointed out, we revised it again in the order of explanation of the quality of life, impact of quality of life, and variables of care providers in nursing hospitals. |
|
3. Try to provide a more comprehensive picture of global findings in this venue - it is a subject that is well studied around the world and can be reported and covered from various angles throughout the world, for example:
Winzelberg, G. S., Williams, C. S., Preisser, J. S., Zimmerman, S., & Sloane, P. D. (2005). Factors associated with nursing assistant quality-of-life ratings for residents with dementia in long-term care facilities. The Gerontologist, 45(suppl_1), 106-114 Bowers, B. J., Esmond, S., & Jacobson, N. (2000). The relationship between staffing and quality in long-term care facilities: Exploring the views of nurse aides. Journal of nursing care quality, 14(4), 55-64 Shin, J. H., & Bae, S. H. (2012). Nurse staffing, quality of care, and quality of life in US nursing homes, 1996–2011: an integrative review. Journal of gerontological nursing, 38(12), 46-53 |
I added and revised the sentence based on the reference you gave me. Thank you. ->On the other hand, although there exists a correlation between fatigue, depression, job stress, and quality of life, the study did not find these factors to be direct determinants of caregivers' quality of life. The quality of life for caregivers is influenced by various factors, including the severity of the disease in dementia patients admitted to nursing homes, the caregiver's perception of dementia, and family of the patients’ care needs [57-1]. A higher number of employees correlates with a higher quality of life for caregivers [57-2,57-3], suggesting a positive impact on the overall quality of service. In essence, the number of caregivers, the severity of dementia in patients, and family of the patients’ care needs collectively affect the caregiver's quality of life. These factors can serve as significant prerequisites for caregivers to experience fatigue, depression, and job stress in the performance of their duties. Surprisingly, in this study, fatigue, depression, and job stress did not emerge as major influencing factors, presenting results that differ from previous studies. Consequently, it is imperative to conduct further studies, taking into account these aforementioned factors in future research. Such investigations are crucial for a comprehensive understanding of the dynamics affecting caregiver well-being.
|
|
4. In the sample selection - the criteria for selection is unclear - why only women aged 41 and above, for example?
|
We will revise the criteria for selecting subjects to adult women. |
|
5. In the procedure there is no point in assigning code names (city A city B etc.) - either provide the actual city names or just say " a sample taken from 6 different cities in S. Korea" that can be enough. |
Corrected the sentence as the reviewer pointed out. -> As for the sample size, 137 female caregivers were conveniently extracted from 9 nursing hospitals in 6 cities located in the central region of Korea. |
|
6. Data analyses and results: I see little point in dividing variables values into categorical data. What is the meaning the answers yes or no to religion? which religion? what level of religiosity? same goes with age or work experience - leave the numbers (age in years, experience in years) as they are and gain more statistical power in your regression results!!
|
In Korea, the Institutional Bioethics Committee must ask whether they have religion or not because of the problem of collecting personal information to approve it. |
|
7. I see no added value in a series of ANOVA or t-tests - step-wise regression in which demographics are entered in the first step, then the main variables are entered will provide the same data with more information about the nature of the effects found. |
Thank you for your guidance. |
|
8. The discussion is decent. Please refrain from any language suggesting causation: speak of associations, explained variance but not influence since you utilized a correlational study design which does not allow the establishing of causal relationships between variables. Develop the limitations section, refer not only to the sample, but also to the potential intervening nature of a specific culture, the use of specific questionnaires, etc.
|
We mentioned limitations and cultural factors, and added a description of variables that were correlated but had no effect. ->As a limitation, care should be taken when interpreting the results because this study focused on specific areas in the central region. In this study, only those who could communicate in Korean were selected as subjects. Therefore, it is suggested that the research be expanded in the future, including foreign caregivers working in nursing hospitals. At this time, we need the help of a foreign leader who can interpret. -> On the other hand, although there exists a correlation between fatigue, depression, job stress, and quality of life, the study did not find these factors to be direct determinants of caregivers' quality of life. The quality of life for caregivers is influenced by various factors, including the severity of the disease in dementia patients admitted to nursing homes, the caregiver's perception of dementia, and family of the patients’ care needs [54]. A higher number of employees correlates with a higher quality of life for caregivers [55,56], suggesting a positive impact on the overall quality of service. In essence, the number of caregivers, the severity of dementia in patients, and family of the patients’ care needs collectively affect the caregiver's quality of life. These factors can serve as significant prerequisites for caregivers to experience fatigue, depression, and job stress in the performance of their duties. Surprisingly, in this study, fatigue, depression, and job stress did not emerge as major influencing factors, presenting results that differ from previous studies. Consequently, it is imperative to conduct further studies, taking into account these aforementioned factors in future research. Such investigations are crucial for a comprehensive understanding of the dynamics affecting caregiver well-being.
|

Reviewer 4 Report (New Reviewer)
Comments and Suggestions for Authors
In the abstract you list "fatigue, depression, self-efficacy, job stress, interpersonal relationship, quality of life" as the study variables, then in the next sentence when describing results also talk about education, income, hobbies, and disease (while not mentioning fatigue, stress, or self efficacy). Be consistent with study variables.
In the introduction, the term "women's work" is a little loaded, I'd suggest revising this statement.
There is a focus on fatigue, depression, and stress in the introduction, yet these are absent from the results.
In the materials, it is unclear what "middle aged" and "elderly" mean - what is the distinction?
Under inclusion criteria - what does "those that can communicate" mean?
What was the data for the sample size calculation? Was it specific to the impact of fatigue etc. on quality of life? A priori power analysis has to be based on the variables of interest or the effect size is meaningless for the current study.
Describing the self-efficacy tool, you say "in Shin's study..." but the lead author is Chen.
Describing the job stress tool, you say "in Kwan's study..." however this researcher is not mentioned in the development or adaptation of the tool.
It is unclear in the analysis section why hierarchical models are employed - what is the rational? What are the levels of unit analysis? Why not just linear regression?
Were questionnaires given to participants on paper? If so, how were they inputed to digital for analysis? Where was this data stored?
The table 1 variables are all different from the study variables of interest described in the methods section?
In your description of correlation results, you have fatigue at r=0.3, which is a positive correlation (more fatigue = higher QOL) - is this correct? Or is the number reported in the table correct?
In section 4.5 - you are treating the demographic variables as key predictor variables, instead of control variables. You don't talk about fatigue, stress, or depression effects. Does that mean they weren't significant? This needs to be presented and discussed (beyond just table 4). This is the main result of your paper - that fatigue, stress, and depression did not predict quality of life in your participants.
You start your discussion with "This study aimed to determine the impact of fatigue, depression, self-efficacy, job 360 stress, and interpersonal relationships on the quality of life (QoL)" and yet you don't actually talk about these, just your demographic variables.
You then say that QOL if influenced by fatigue - but this was insignificant in the regression model.
Comments on the Quality of English Language
Some editing for language required.
Author Response
|
Reviewer 4 |
Thank you for your sincere review opinion. I revised it hard to reflect your review. |
|
1.In the abstract you list "fatigue, depression, self-efficacy, job stress, interpersonal relationship, quality of life" as the study variables, then in the next sentence when describing results also talk about education, income, hobbies, and disease (while not mentioning fatigue, stress, or self efficacy). Be consistent with study variables. |
Revised and added the sentence as the reviewer pointed out.
The purpose of this study was to analyze the effects of general characteristics, fatigue, depression, self-efficacy, job stress and interpersonal relationships on the quality of life (QoL) of caregivers in nursing hospitals and use them as basic data for intervention programs to improve the quality of life of caregivers. Since the subject's fatigue, depression, and stress did not affect the quality of life, further research is needed. |
|
2.In the introduction, the term "women's work" is a little loaded, I'd suggest revising this statement.
|
Changed sentences -> In modern society, women's job performance goes beyond just economic necessity and represents a means of self-realization. Therefore, working life can be a source of psychological satisfaction and stress and has a profound effect on physical and mental health. |
|
3.There is a focus on fatigue, depression, and stress in the introduction, yet these are absent from the results.
|
Theoretically, fatigue, depression, and stress were included as variables because it was determined that they were related to the quality of life, but it was found that there was no relevance and influence on the quality of life of the subjects in this study. Therefore, finding out that there is no relevance is also considered meaningful as a result of the study for Korean caregivers, especially caregivers working in nursing hospitals. I added this to the discussion. Thank you for pointing it out. There is a discussion about the correlation, but there was no significant result in the influencing factors, so I added this part to the discussion as follows. ->On the other hand, although there exists a correlation between fatigue, depression, job stress, and quality of life, the study did not find these factors to be direct determinants of caregivers' quality of life. The quality of life for caregivers is influenced by various factors, including the severity of the disease in dementia patients admitted to nursing homes, the caregiver's perception of dementia, and family of the patients’ care needs [57-1]. A higher number of employees correlates with a higher quality of life for caregivers [57-2,57-3], suggesting a positive impact on the overall quality of service. In essence, the number of caregivers, the severity of dementia in patients, and family of the patients’ care needs collectively affect the caregiver's quality of life. These factors can serve as significant prerequisites for caregivers to experience fatigue, depression, and job stress in the performance of their duties. Surprisingly, in this study, fatigue, depression, and job stress did not emerge as major influencing factors, presenting results that differ from previous studies. Consequently, it is imperative to conduct further studies, taking into account these aforementioned factors in future research. Such investigations are crucial for a comprehensive understanding of the dynamics affecting caregiver well-being. |
|
4.In the materials, it is unclear what "middle aged" and "elderly" mean - what is the distinction?
|
As already mentioned, it is peculiar that most of the caregivers working in nursing hospitals in Korea are middle-aged women over 41 and elderly women over 65. This is a very unusual phenomenon, especially considering that caregivers in acute hospitals are young adult women. |
|
5.Under inclusion criteria - what does "those that can communicate" mean?
|
As mentioned in the study's limitations, there are many Korean-Chinese women working in nursing hospitals. These are women who do not fully understand and interpret the questionnaire. Therefore I changed the questionnaire to someone who can understand and fill it out ->5) Those who can understand the questionnaire and fill it out.
|
|
6.What was the data for the sample size calculation? Was it specific to the impact of fatigue etc. on quality of life? A priori power analysis has to be based on the variables of interest or the effect size is meaningless for the current study. |
In order to measure the sample size, a preliminary study was presented as a reference, and the G-power 3.1.9.7. Program, a representative sample size measurement program, was used. First of all, the independent variables were correlated with the quality of life, which is the dependent variable, and the sample size was measured, including variables that showed differences according to general characteristics. And it is said that most correlation or regression studies in Korea are based on the middle size of 0.15. |
|
7.Describing the self-efficacy tool, you say "in Shin's study..." but the lead author is Chen. |
Changed -> Chen et al.’s study [38], the reliability was Cronbach's α=.87 |
|
8.Describing the job stress tool, you say "in Kwan's study..." however this researcher is not mentioned in the development or adaptation of the tool. |
Changed This study presented the reliability of Lee & Park's tool because it used a tool in Lee & Park's research, a tool modified to a Korean style, not a tool at the time of development.->. In Lee & Park's study [41], the reliability by half reliability was .85, |
|
9.It is unclear in the analysis section why hierarchical models are employed - what is the rational? What are the levels of unit analysis? Why not just linear regression? |
Revised ->As a method of analyzing factors affecting the QoL of care providers, first, the influencing factors among the general characteristics of the subject were identified, followed by factors such as biological function, functional status, and general health perception as the overall health response of care providers, and then environmental factors and variables that can provide mediating effects were analyzed. Therefore, a hierarchical analysis was performed to analyze the results and explanatory power of factors affecting quality of life through this hierarchical analysis. |
|
10.Were questionnaires given to participants on paper? If so, how were they inputed to digital for analysis? Where was this data stored? |
Revised ->Data collection was collected after distributing a paper questionnaire to the subject, coded in Excel to the computer, and called to SPSS Statistics 27.0 program for statistical analysis. After that, the coded data is stored in a file with a password on the computer owned by the lead researcher in accordance with the principles of the Institutional Bioethics Committee, making it difficult for others to access it. And keep the paper questionnaire in a bookshelf with a lock. After writing the paper, the file will be stored on the computer and the paper questionnaire will be stored on the bookshelf for three years, and the file will be deleted in a way that cannot be recovered and the paper questionnaire will be crushed with a shredder. |
|
11.The table 1 variables are all different from the study variables of interest described in the methods section? |
In Table 1, the difference in quality of life according to general characteristics and general characteristics was tested, and significant variables were included as control variables and used for hierarchical regression analysis. Overall, general characteristics and independent variables were used to understand situations in which explanatory power was added to the quality of life, which is a dependent variable. Thus, Table 1 is an important result of the study. |
|
12.In your description of correlation results, you have fatigue at r=0.3, which is a positive correlation (more fatigue = higher QOL) - is this correct? Or is the number reported in the table correct? |
r=-.30 (p<.001) is correct in Table 3. We modified r=.30 to r=-0.3 in the text. |
|
13.In section 4.5 - you are treating the demographic variables as key predictor variables, instead of control variables. You don't talk about fatigue, stress, or depression effects. Does that mean they weren't significant? This needs to be presented and discussed (beyond just table 4). This is the main result of your paper - that fatigue, stress, and depression did not predict quality of life in your participants. |
Variables showing differences in quality of life according to general characteristics were analyzed in the first stage by considering them as control variables. |
|
14.You start your discussion with "This study aimed to determine the impact of fatigue, depression, self-efficacy, job stress, and interpersonal relationships on the quality of life (QoL)" and yet you don't actually talk about these, just your demographic variables. |
In the discussion, independent variables that have a significant influence were explained, and variables such as fatue, stress, and depression, which were not identified as influencing factors, were also added. On the other hand, although there exists a correlation between fatigue, depression, job stress, and quality of life, the study did not find these factors to be direct determinants of caregivers' quality of life. The quality of life for caregivers is influenced by various factors, including the severity of the disease in dementia patients admitted to nursing homes, the caregiver's perception of dementia, and family of the patients’ care needs [54]. A higher number of employees correlates with a higher quality of life for caregivers [55,56], suggesting a positive impact on the overall quality of service. In essence, the number of caregivers, the severity of dementia in patients, and family of the patients’ care needs collectively affect the caregiver's quality of life. These factors can serve as significant prerequisites for caregivers to experience fatigue, depression, and job stress in the performance of their duties. Surprisingly, in this study, fatigue, depression, and job stress did not emerge as major influencing factors, presenting results that differ from previous studies. Consequently, it is imperative to conduct further studies, taking into account these aforementioned factors in future research. Such investigations are crucial for a comprehensive understanding of the dynamics affecting caregiver well-being. |
|
15.You then say that QOL if influenced by fatigue - but this was insignificant in the regression model. |
Fatigue was not identified as an influencing factor, so we corrected the wrong explanation. |
Round 2
Reviewer 4 Report (New Reviewer)
Comments and Suggestions for Authors
Thank you for addressing my concerns. I still think in the abstract and introduction you should clarify age of "middle aged" and "elderly". I also think in the inclusion criteria, with the item "able to understand the questionnaire" specify whether this means fluency in the language. Overall, the paper is much improved.
Comments on the Quality of English LanguageThere are some concerns still about English language (sentence structure etc.) however overall the paper is much clearer.
Author Response
|
review |
revision |
|
Thank you for addressing my concerns. I still think in the abstract and introduction you should clarify age of "middle aged" and "elderly". I also think in the inclusion criteria, with the item "able to understand the questionnaire" specify whether this means fluency in the language. Overall, the paper is much improved. |
Thank you very much for guiding me in crafting a strong thesis. We have addressed the age-related and comprehension issues in the questionnaire as per your suggestions. The abstract, introduction, and body have been revised accordingly. Thank you. |
|
|
In abstract Methods: The participants in the study were 137 caregivers, aged 52-76, who were actively working in nursing hospitals. In introduction Most caregivers in Korean nursing hospitals are middle-aged women over 41 and elderly women over 65, who are not well-off at home. Given that caregivers, predominantly middle-aged women over 41 and elderly women over 65, exhibit physical, psychological, and social characteristics, they are particularly sensitive to fatigue [14]. 3.1. Participants Participants included middle-aged women (41 years and older) and elderly women (65 years and older) who served as caregivers in nursing hospitals. 5) Individuals who are fluent in Korean and can comprehend and complete the questionnaire. 5.Discussion In particular, middle-aged women over 41 who simultaneously manage work and household chores were found to experience increased fatigue and stress [27]. 6. Conclusions The purpose of this study was to identify the factors affecting the quality of life of middle-aged women over 41 and elderly women over 65 serving as caregivers in nursing hospitals. Self-efficacy and interpersonal relationships are crucial factors for enhancing the quality of life for middle-aged women over 41 and elderly women over 65 serving as caregivers in nursing hospitals. |
This manuscript is a resubmission of an earlier submission. The following is a list of the peer review reports and author responses from that submission.
Round 1
Reviewer 1 Report
Comments and Suggestions for Authors
Raw 18: what means stage 1 and further on stage 2? They are the steps of analysis (the univariate correlation?) or have another meaning?
The introduction is quite long and should be better synthetized. It looks like a list of items, more than a brief of the current knowledge, with notions that are not explained or conclusion which are not sustained. I will briefly name some example but in my opinion the whole introduction should be re-formulated.
· Raw 49: what means” job dependency”?
· Raws 61-68: if the quality of life is “the interplay of social conditions, systems, societal member relationships”, how could it raise to the conclusion that the “medical professionals (which are those?) must make efforts to enhance the quality of life of caregivers”. It seams more like the organizers of the system or of the society should be in charge.
· Raw 87-89: is this a gender specific characteristic?
· Raw 116: fatigue is generally considered a symptom
· Raw 118: depression is not a symptom, but a disease
· Raw: 121: job related stress is a perception of the work content and context. Stressors at work are the risk factors or “environmental characteristics”, as some of them are presented in Table 2
Methods:
Raw 127 and 128 repeat the same things. This part should be re-formulated.
Raw 139: what means “those who were capable of responding to the questionnaire and communicating efficiently?”. The questionnaire was so difficult that people should need special training? Are there caregiveres who are not able to communicate? If so, how can they do their job? This criteria raises several biases to study.
Raw 190: the items were excluded by Lee and Park or by the authors of this article?
Raw 204-214: if the outcome has similar components with the factors of influence (e.g. the scale for fatigue and for the quality of life) the results have no value.
The analysis presented in the paragraph “Differences in QoL according to general characteristics of participants “ have almost the same conclusionsas the one described in “factors affecting the QoL of the participants” (which is obvious, as there are the same values). The last one is more significant as it tests also the influence of different variables. In my opinion the paragraph “Differences in QoL according to general characteristics of participants” including Table 1 is redundant and should be excluded. Table 1 might be replaced with a general description of the study population and characteristics.
As mentioned previosuly, if the items of the questionnaire on fatigue or depression are similar with items included in the QoL questionnaire the relation identified is totally irelevant.
After clarifying these issues, the discussion section should be revised accordingly.
Comments on the Quality of English LanguageThe quality of english is average, a revision by a native english speaker would be preferable.
Author Response
|
|
Reviewer 1 |
Thank you for your sincere review opinion. We revised it hard to reflect your review. |
|
1 |
Raw 18: what means stage 1 and further on stage 2? They are the steps of analysis (the univariate correlation?) or have another meaning?
|
The analysis method of this study used a hierarchical regression analysis method. The reason was to identify and interpret the results of influencing factors, including variables that are correlated and thought to affect step by step. Therefore, the first-stage regression analysis was analyzed including variables related to general characteristics, and as a result, economic status, hobbies, and leisure activities were identified as influencing factors. In the second stage, as a result of additional analysis of independent variables, that is, fatigue and depression, which are believed to affect quality of life, monthly income and perceived economic status were found to be significant influencing factors among general characteristics. Therefore, it is intended to interpret its meaning by analyzing it in a hierarchical manner during regression analysis. |
|
|
The introduction is quite long and should be better synthetized. It looks like a list of items, more than a brief of the current knowledge, with notions that are not explained or conclusion which are not sustained. I will briefly name some example but in my opinion the whole introduction should be re-formulated. ·
|
As the reviewer pointed out, a lot of content was added to the introduction, focusing on the concepts used in the study. Thank you. Ex. However, there is a lot of job-related stress because of unsafe working conditions/environments, heavy workload, scarcity of work, interpersonal conflicts at work, role ambiguity, job insecurity and bi-dayly work , members of workforce can feel uncomfortable, pressured, tense, and conflicted. According to previous studies, nursing caregivers experience a high quality of life when providing care for vulnerable elderly individuals in nursing hospitals. This is achieved through accurately identifying the needs of patients, delivering appropriate care, and effectively managing various situations [16]. Therefore, improving the caregiving skills of caregivers, fostering positive interrelationships, and enhancing environmental conditions in the workplace can potentially enhance the quality of life for caregivers. Firstly, fatigue in office workers is typically caused by excessive mental and physical labor. In the case of nursing caregivers, the inherently labor-intensive nature of their work targeting humans adds both physical and mental fatigue, posing a threat to their health [19-1] (Osaki et al., 2016). Given that caregivers are predominantly middle-aged or elderly women, the physical, psychological, and social characteristics of this demographic make them particularly sensitive to fatigue [19-2]. (Powell et al., 2002) Moreover, the repetitive physical tasks involved in caregiving, such as changing body positions, assisting with dietary needs, replacing diapers, bathing, and providing mobility assistance in nursing hospitals, contribute to a heightened perception of fatigue. Caregivers also reported rapid fatigue due to the need to suppress individual emotions and maintain constant emotional control for kindness and patience in interactions with patients, families, and medical staff (Kim, 2014). Notably, factors such as sleep disorders and the demands of shift work in nursing hospitals were identified as contributors to increased fatigue. When caregivers experience emotional exhaustion while performing their duties, it can lead to emotional states that pose a threat to mental health, such as anxiety and depression, ultimately resulting in a decline in the quality of life Meanwhile, self-efficacy is a concept related to the evaluation of subjective cognitive abilities to successfully perform tasks and serves as a key factor in human achievement [19-5] (Bandura, 1993). In other words, an individual's belief in themselves influences their behavior, and the level of their behavior varies depending on how well they perceive their ability to perform the task. In addition, humans grow and develop within various human relationships, with these connections serving as a key factor in determining the quality of human life.. Workplace relationships, in particular, significantly impact job satisfaction and overall quality of life |
|
|
Raw 49: what means” job dependency”?
· Raws 61-68: if the quality of life is “the interplay of social conditions, systems, societal member relationships”, how could it raise to the conclusion that the “medical professionals (which are those?) must make efforts to enhance the quality of life of caregivers”. It seams more like the organizers of the system or of the society should be in charge. · Raw 87-89: is this a gender specific characteristic? · Raw 116: fatigue is generally considered a symptom · Raw 118: depression is not a symptom, but a disease · Raw: 121: job related stress is a perception of the work content and context. Stressors at work are the risk factors or “environmental characteristics”, as some of them are presented in Table 2
|
Raw 49; changed the sentence However, there is a lot of job-related stress because of the heavy workload of caregivers and working with employees in various occupations, the scope of work is ambiguous and most work methods are bi-day [3,6].
Raw 61-68 As you pointed out, the meaning was not correct, so I deleted it and described it again. Ultimately, the quality of life results from the interplay of social conditions, systems, and societal member relationships, representing overall satisfaction that makes an individual's life valuable and rewarding [14->15]. Hence, the quality of life of caregivers can improve when they are satisfied with their job while providing care services in nursing hospitals [15->16]. According to previous studies, nursing caregivers experience a high quality of life when providing care for vulnerable elderly individuals in nursing hospitals. This is achieved through accurately identifying the needs of patients, delivering appropriate care, and effectively managing various situations [16->17]. Therefore, improving the caregiving skills of caregivers, fostering positive interrelationships, and enhancing environmental conditions in the workplace can potentially enhance the quality of life for caregivers.
Raw 87-89: Yes, that’s right. This characteristic is higher for female workers than for men. Raw 116, 118: Fatigue was viewed as a health response in response to physical and mental health, and depression was viewed as an emotional state, a matter of mood, and a human response before going to psychotic depression.
Raw 121: Job-related stress is about work content and context. Since it is job-related stress due to problems such as role conflict, role ambiguity, and role overload, it is necessary to devise intervention measures to change these characteristics by designating them as environmental characteristics surrounding caregivers. |
|
|
Methods: Raw 127 and 128 repeat the same things. This part should be re-formulated. Raw 139: what means “those who were capable of responding to the questionnaire and communicating efficiently?”. The questionnaire was so difficult that people should need special training? Are there caregiveres who are not able to communicate? If so, how can they do their job? This criteria raises several biases to study. Raw 190: the items were excluded by Lee and Park or by the authors of this article? |
Raw 127-128 As you pointed out, changed it. The participants are middle-aged or elderly female caregivers in nursing hospitals. There are about 40,000 caregivers working in nursing hospitals nationwide. The sample size was 137 patients from 9 nursing hospitals located in the central region, and they were specifically conveniently sampled one in County B, one in City C, two in City D, three in City G, one in City J, and one in City N.
Raw 139: changed it Those who can communicate. . Raw 190: changed it Researchers deleted three items with low reliability and validity. Finally, the tool consists of 11 items. |
|
|
Raw 204-214: if the outcome has similar components with the factors of influence (e.g. the scale for fatigue and for the quality of life) the results have no value. The analysis presented in the paragraph “Differences in QoL according to general characteristics of participants “ have almost the same conclusionsas the one described in “factors affecting the QoL of the participants” (which is obvious, as there are the same values). The last one is more significant as it tests also the influence of different variables.
|
Rae 204-214 Right, if the tool of the dependent variable and the variable of the independent variable are the same, then it's meaningless. However, in this study, the area of quality of life refers to the area in a large frame, that is, the physical, mental, social, and environmental area, and I don't think the variable selected as an independent variable is like poison. For example, depression does not include the entire mental domain, but the concept of depression will affect the quality of life through a literature survey, and it was analyzed for middle-aged or elderly caregivers in nursing hospitals. So it's hard to agree that it's the same thing. The realm of quality of life itself is so comprehensive that all studies can be interpreted as useless. Please be kind to me. And, in the analysis of influencing factors, it is difficult to include other variables because there must be a correlation between variables before analysis. |
|
|
In my opinion the paragraph “Differences in QoL according to general characteristics of participants” including Table 1 is redundant and should be excluded. Table 1 might be replaced with a general description of the study population and characteristics. |
Table 1: The number and percentage of general characteristics in Table 1 are shown, and the results of quality of life according to general characteristics are presented at once so that the same table is not drawn again. |
|
|
As mentioned previosuly, if the items of the questionnaire on fatigue or depression are similar with items included in the QoL questionnaire the relation identified is totally irelevant. After clarifying these issues, the discussion section should be revised accordingly.
Comments on the Quality of English Language The quality of english is average, a revision by a native english speaker would be preferable. |
It is not the same as the questionnaire of independent variables such as fatigue and depression and the question of quality of life. Therefore, there is no problem in using it as a research variable. |
Reviewer 2 Report
Comments and Suggestions for Authors
The article is quite well thought out, both in its introduction and methodology. I consider that the discussion part should be improved by comparing the results with a larger number of studies than the ones used.
Author Response
|
Reviewer 2 |
Thank you for your sincere review opinion. We revised it hard to reflect your review. |
|
The article is quite well thought out, both in its introduction and methodology. I consider that the discussion part should be improved by comparing the results with a larger number of studies than the ones used. |
We supplemented the discussion by adding references. Thank you. In terms of education and perceived economic status, individuals with a college degree or higher experienced a higher quality of life compared to those with only a high school diploma. Similarly, those with a monthly income of $1500 or more enjoyed a higher quality of life than those earning less than $1500. Furthermore, the group that received higher education demonstrated a higher quality of life than the group that did not, and those engaged in hobbies and leisure activities reported a higher quality of life than those who were not involved in such activities. These findings underscore the correlation between education, economic factors, and overall quality of life. The higher quality of life among individuals with advanced education is attributed to their elevated economic achievement and the diverse opportunities for life activities available to them [6,23]. Monthly income and economic status play a pivotal role in enabling individuals to engage in hobbies and leisure activities, aligning with the findings that those with such pursuits experience a higher quality of life compared to those without [8]. Therefore, to enhance the quality of life for care providers in nursing hospitals and foster satisfaction in their work, it is imperative for the individuals overseeing these institutions to develop educational programs and ensure continuous education. Additionally, recognizing the significance of economic aspects, it is crucial to provide caregivers with appropriate compensation. Measures should be devised to establish clubs for hobbies and leisure activities, along with allowances to promote the overall welfare of the employees. In addition, individuals who are married or report high marital satisfaction exhibit a higher quality of life compared to those experiencing bereavement, divorce, or dissatisfaction with marriage. This quality of life is also higher than that observed in disease-free cases. The marital satisfaction of middle-aged women has been found to be positively correlated with their quality of life, contributing to psychological stability [48]. Marriage status and marital satisfaction serve as sources of social support, encompassing assistance from both spouses and family. Spousal support emerges as a crucial variable for maintaining psychological stability and overall health [48]. Similarly, family consideration and assistance constitute support systems that aid caregivers in effectively managing their roles in the workplace [6]. In instances of unsatisfactory marriages, emotional crises, such as helplessness, despair, and unhappiness, are commonly experienced. Thus, there is a need to formulate strategies aimed at increasing social support to enhance the quality of life for nursing care workers. Furthermore, it aligns with reported findings that the presence of a disease leads to decreased work efficiency, deterioration in health, prolonged return-to-work periods, and ultimately lower quality of life for workers [47]. Consequently, it is essential to make concerted efforts to maintain and promote the health of caregivers themselves. This includes focusing on disease prevention, regular health check-ups, and the steadfast maintenance of healthy lifestyles in their daily lives. Fatigue is an empirical phenomenon that leads to a decrease in the ability to perform daily activities when the burden increases due to excessive mental and physical exertion. It serves as a sign of homeostasis disruption and contributes to a decline in physical, mental, and emotional capabilities. Studies have reported that stress and depression are risk factors associated with this fatigue [53]. Middle-aged working women, in particular, experience higher levels of job stress compared to their counterparts [54]. This heightened stress, coupled with physical and psychological burdens, results in fatigue, ultimately leading to a diminished quality of life [23, 55]. In addition, Kwon's results [56] indicate that the quality of life of nurses working in hospitals is inversely proportional to the levels of depression and job stress. Notably, middle-aged and elderly working women, who juggle both work and housework, experience heightened fatigue and stress [31]. This is particularly true for shift workers, as fatigue persists even after sleep. If not addressed, persistent fatigue can pose serious threats to physical and mental health, leading to issues such as overwork, reduced productivity, cardiovascular problems, cancer, and depression [57]. Given these challenges, for caregivers compelled to work shifts due to the nature of their job, it is imperative to implement measures aimed at reducing fatigue, job stress, and depression. On the other hand, although there exists a correlation between fatigue, depression, job stress, and quality of life, the study did not find these factors to be direct determinants of caregivers' quality of life. Therefore, it is recommended to expand the sample size in future studies and conduct repeated investigations that include these concepts. In traditional Korean culture, the basic axis of social relations is the 'ourist group'.In other words, in the social relationship of Koreans, an individual is not independent, but rather a 'relationship individual' who becomes one with another called 'we'. Koreans establish an interpersonal relationship framework within our category, focusing on school ties, regional ties, and blood ties, believe that it is a safe relationship, thoroughly preserve and strengthen the relationship, unconditionally accept it, and have full trust [64].
references 7 Park, J.O. The Empirical Study on the Preventive Management Strategy of the Job Stress. Ph.D. Thesis, Hannam University, Daejeon, Korea, 2008. 21 Osaki, T., Morikawa, T., Kajita, H., Kobayashi, N., Kondo, K., & Maeda, K. Caregiver burden and fatigue in caregivers of people with dementia: Measuring Human Herpesvirus (HHV)-6 and-7 DNA levels in saliva. Archives of Gerontology and Geriatrics, 2016, 66, 42-48. https://doi.org/10.1016/j.archger.2016.04.015 22 Powell, L. H., Lovallo, W. R., Matthews, K. A., Meyer, P., Midgley, A. R., Baum, A., et al. Physiologic markers of chronic stress in premenopausal, middle-aged women. Psychosomatic Medicine 2002, 64(3), 502-509. https://doi.org/10.1097/00006842-200205000-00015 24 Kim, H. S. The effects of emotional labor, long term care service self-efficacy of geriatric hospital workers on their job satisfaction. Unpublished master's thesis, Konyang University, Daejeon, 2014. 25 Kim, J. Y.; Lee, J. M. A Study on the psychological well-being of nursing care workers. Journal of Dong-A Humanities 2017, 12, 383-420. 26 Bandura, A. Perceived Self-Efficacy in Cognitive Development and Functioning. Educational Psychologist, 1993, 28, 117-148. http://dx.doi.org/10.1207/s15326985ep2802_3 29 Song, M. J.; Lee, H. R. The Development and Effects of A Human Relationship Training Program Using a University Liberal-Art Course. The Korean Journal of Counseling and Psychotherapy 2008, 20(2), 269-291 48. Yeo, J. H. Correlational Study on Managemen t of Menopause, Marittal S atisfaction , and Quality of Life in Middle-aged Wom en. J Korean Acad Nurs 2004, 34(2), 261-269. 53 Kim, H. K. Fatigue and Factors Influencing Fatigue in Middle-aged Adults by Age Groups. Korean J Women Health Nurs 2006, 12(4), 273-281. 54 Griffiths, A.; Knight, A.; Mahudin, A. M.; Diana, N. Ageing, work-related stress and health: Reviewing the evidence. A Report for Age Concern and Help the Aged and TAEN (The Age and Employment Network. London). 2009. 55 Park, J. W.; Kwon, M. J. The Effect of Well-being, Fatigue, and Self-efficacy on Health Promotion Behaviors among Shift Workers. Korean Journal of Occupational Health Nursing Vol. 28 No. 4, 293-299. https://doi.org/10.5807/kjohn.2019.28.4.293 57 Park, S. P.; Lee, D. B.; Kwon, I. S.; Cho, Y. C. Analysis of the influence of job stress and psychosocial factors on self per[1]ceived fatigue in white collar male workers using the struc[1]tural equation model. Annals of Occupational and Environmental Medicine 2010, 22(1), 48-57. 64 Cha, Y. S. Characteristics and Problems of Korean Interpersonal Culture. Gyeryongilbo, 19 December 2014. (assessed by 18 November 2023). http://www.gyeryongilbo.com/news/articleView.html?idxno=14026
|

Reviewer 3 Report
Comments and Suggestions for Authors
Kim and Oh conducted a cross-sectional study to analyze the effects of fatigue, depression, self-efficacy, job stress, and interpersonal relationships on the quality of life (QoL) of caregivers in nursing hospitals. They found that factors such as age, marital status, marital satisfaction, education level, monthly income, economic status, engagement in hobbies or leisure activities, and the number of diseases had an impact on QoL.
The study is relevant for informing interventions aimed at improving the QoL of caregivers; however, there are several issues that have diminished my enthusiasm for it.
The title is excessively long and confusing. I suggest a more concise title like "Determinants of Quality of Life in Female Caregivers in Elderly Care Facilities."
The study is limited by being cross-sectional and non-representative, which reduces the generalizability of the results. It only included middle-aged and older women, leaving the question of why only women were included. The methods state "female adults aged 41 and above," but the table indicates that only females aged 52 or above were considered.
The sample size was determined based on predictor variables, but there is no information about the representativeness of the sample in relation to the population studied. How big is the population studied?
The analyses were not adjusted for multiple comparisons, and the assumption of normality was made but not verified.
The description of the regression analysis is insufficient. It appears that dummy variables were created, but no information is provided about them. Redundant and contradictory variables, such as economic status (high and low) and monthly income, were included.
In the discussion, the authors stated that "For Koreans, interpersonal relationships form the foundation of emotions and hold absolute importance in life." However, it is not clear whether this specificity applies only to Koreans or if there are references to justify this claim.
Minor corrections include fixing the word "gun" in the abstract (possibly meaning "county") and revising the use of "<" and ">" symbols in tables.
Comments on the Quality of English LanguageI stated my suggestions above.
Author Response
|
|
Reviewer 3 |
Thank you for your sincere review opinion. We revised it hard to reflect your review. |
|
1 |
Kim and Oh conducted a cross-sectional study to analyze the effects of fatigue, depression, self-efficacy, job stress, and interpersonal relationships on the quality of life (QoL) of caregivers in nursing hospitals. They found that factors such as age, marital status, marital satisfaction, education level, monthly income, economic status, engagement in hobbies or leisure activities, and the number of diseases had an impact on QoL. The study is relevant for informing interventions aimed at improving the QoL of caregivers; however, there are several issues that have diminished my enthusiasm for it. |
Thank you very much for your careful review so that it can be a good thesis. As you pointed out, I revised it hard. Thank you. |
|
2 |
The title is excessively long and confusing. I suggest a more concise title like "Determinants of Quality of Life in Female Caregivers in Elderly Care Facilities." |
changed to the title you recommended Determinants of Quality of Life in Female Caregivers in Elderly Care Facilities |
|
3 |
The study is limited by being cross-sectional and non-representative, which reduces the generalizability of the results. It only included middle-aged and older women, leaving the question of why only women were included. The methods state "female adults aged 41 and above," but the table indicates that only females aged 52 or above were considered. |
What the reviewer pointed out is the reality of care providers working in nursing hospitals in Korea. Most of the care providers in nursing hospitals are women, mainly in middle and old age, and visited many nursing hospitals in the central region to collect data, but women in their 50s and older were working in middle age. In discussion, added as below. ->And since the subjects of this study were women in their 50s or older in middle age, it is suggested that future studies recruit and study subjects in consideration of the age group. |
|
4 |
The sample size was determined based on predictor variables, but there is no information about the representativeness of the sample in relation to the population studied. How big is the population studied? |
Added population of caregivers There are about 40,000 caregivers working in nursing hospitals nationwide. |
|
5 |
The analyses were not adjusted for multiple comparisons, and the assumption of normality was made but not verified. |
To identify the problem of multicollinearity, we presented and interpreted the values of Durbin Watson, p-p plot, range of tolerance, and VIF. |
|
6 |
The description of the regression analysis is insufficient. It appears that dummy variables were created, but no information is provided about them. Redundant and contradictory variables, such as economic status (high and low) and monthly income, were included. |
The contents of dummy treatment were described to the text as follows. In the first step of the hierarchical regression analysis, variables that showed differences in QoL based on the general characteristics of the participants, such as age, marital status, marital satisfaction, education, educational experience for improving QoL, monthly income, economic status, and hobbies or leisure activities, were included in the analysis. These variables underwent dummy coding. Below the table 4, the processing status of dummy variables is presented in detail. reference; age (52-59=0, 60-69=1), marriage (married=0, widowed, divorced, or separated =1), marital satisfaction (unsatisfied = 0, satisfied = 1), education (high school graduated or lower = 0, college graduated or higher = 1), educational experience of QoL (none=0, ≥1 = 1), monthly income (<1500 dollars=0, ≥1500 dollars=1), perceived economic status (high=0, low=1), hobby or leisure activity(don’t have=0, have=1) And in order to eliminate the overlapping problem, the actual monthly income and economic status were expressed in terms of their own perception. -> perceived economic status |
|
7 |
In the discussion, the authors stated that "For Koreans, interpersonal relationships form the foundation of emotions and hold absolute importance in life." However, it is not clear whether this specificity applies only to Koreans or if there are references to justify this claim. |
added the characteristics of Koreans
In traditional Korean culture, the basic axis of social relations is the 'ourist group'. In other words, in the social relationship of Koreans, an individual is not independent, but rather a 'relationship individual' who becomes one with another called 'we'. Koreans establish an interpersonal relationship framework within our category, focusing on school ties, regional ties, and blood ties, believe that it is a safe relationship, thoroughly preserve and strengthen the relationship, unconditionally accept it, and have full trust[]. Cha, Y. S. Characteristics and Problems of Korean Interpersonal Culture. Gyeryongilbo, 19 December 2014. http://www.gyeryongilbo.com/news/ articleView.html?idxno=14026 |
|
8 |
Minor corrections include fixing the word "gun" in the abstract (possibly meaning "county") and revising the use of "<" and ">" symbols in tables. |
changed the gun to county in the abstract in 1 county and 5 cities Revised the use of "<" and ">" symbols in tables. < 5, 5 - <10 |